



# Long-term ice phenology records from East Central Europe

Katalin Takács[1], Zoltán Kern[2], László Pásztor[1]

[1]Institute for Soil Sciences and Agricultural Chemistry, MTA Centre for Agricultural Research, Herman Ottó 15, Budapest, 1022, Hungary

[2]Institute for Geological and Geochemical Research, MTA Research Centre for Astronomy and Earth Sciences, Budaörsi 45, Budapest, 1112, Hungary

*Correspondence to*: Katalin Takács (takacs.katalin@rissac.hu), Zoltán Kern (kern.zoltan@csfk.mta.hu)

**Abstract.** A data set of annual freshwater ice phenology was compiled for the largest river (Danube) and the largest lake (Lake Balaton) in East Central Europe, extending regular river and lake ice monitoring data through the use of historical

observations and documentary records dating back to 1774 AD and 1885 AD, respectively. What becomes clear is that the dates of the first appearance of ice and freeze-up have shifted, arriving 12–30 and 4–13 days later respectively per100 years. Break-up and ice-off have shifted to earlier dates by 7–13 and 9–27 days/100 years, except on Lake Balaton, where the date of break-up has not changed significantly. The data sets represent a great potential resource for (paleo)climatological research thanks to the strong, physically determined link between water and air temperature and the occurrence of freshwater

ice phenomena. The derived centennial records of freshwater cryophenology for Danube and Balaton are readily available for detailed analysis of the temporal trends, large-scale spatial comparison or other climatological purposes. The derived dataset is publicly available via PANGAEA at doi.pangaea.de/10.1594/PANGAEA.881056

## 1 Introduction

Freshwater ice is a major component of the terrestrial cryosphere (Brooks et al., 2013). At higher latitudes, mostly in the cold

and the temperate climate zones, many rivers and lakes are covered partly or fully by ice during the winter or even in autumn and spring (Beltaos and Prowse, 2009; Jensen et al., 2007; Weyhenmeyer et al., 2011). Seasonal ice cover can occur on rivers and lakes as far south as 33°N in North America and 26°N in Eurasia. Therefore, several large rivers and lakes from the world's top 15 – numbers 7 and 11 in the ranking – are affected (Prowse et al., 2007a). Although freshwater ice is characterized by an excessive degree of seasonal variability compared to the other components of the cryosphere, it has a

broad ecological and economic significance (Prowse et al., 2006, 2007a). For instance, river ice dynamics remarkably influences riparian and aquatic vegetation (Lind et al., 2014).

The timing and duration of the occurrence of ice phenomena are sensitive to winter weather conditions, especially to air temperature (Prowse et al., 2007b; Smith, 2000), making ice phenology data a good indicator of long-term climate change and inter-annual variability (Klavins et al., 2009; Sharma et al., 2016). There is great potential for climatic research in the

analysis of long-term freshwater ice observations because direct information on climate properties can be provided not only



about recent trends, but also about the temperature regimes even before the start of instrumental temperature observations (Klavins et al., 2009; Magnuson et al., 1999; Sharma et al., 2016). In spite of the fact that observations of freshwater ice phenomena have a long history (Fujiwhara, 1921; Liljequist, 1941) and freshwater ice conditions have been collected routinely at a large number of water bodies, there are only a few stations that have continuous ice phenology data series

covering more than 100 years (Prowse et al., 2007b). The best known long-term river and lake ice datasets are available for Tornionjoki (Finland) and Lake Suwa (Japan), with these records going back 320 and 570 years, respectively (Sharma et al., 2016). The great potential of long-term freshwater ice observations in (paleo)climatological research was recognized more than 40 years ago (Gray, 1974; Williams, 1970). Reflecting its role as a recognized powerful indicator for winter air temperature changes freeze-up records of Lake Suwa was one of the few proxy records used in the first quantitative

reconstruction of Northern Hemisphere annual mean temperatures (Groveman and Landsberg, 1979).

Owing to the long tradition of ice observations in Hungary, some relatively long, covering more than 240 years, freshwater ice phenology datasets are available for East Central Europe as well. These are also suitable in the detection of long-term trends and climate change research. In this study, an ice phenology dataset was compiled for the largest river (Danube) and the largest lake (Lake Balaton) of East Central Europe by combining historical periodical observations, documentary

evidence, and regular river and lake ice monitoring data, resulting in a relatively continuous, daily record of freshwater ice observation from the 18th century to the present. As a result, the longest and most complete river and lake ice regime dataset for East Central Europe is presented here.

## 2 Data product description

### 2.1 Ice phenology data

Freshwater ice can occur in many forms on rivers and lakes. There are particular differences between the process of ice evolution on lakes, with static ice formation, and rivers, which are dynamic in terms of ice formation (Barry and Gan, 2011). However, the common point is that, after the water cools to below 0°C and is therefore supercooled, frazil ice can develop. The other ice forms are then built up from frazil ice particles (Carlson, 1981). The regular freshwater ice observations include the monitoring of the following ice phenomena:

(1) Border ice: ice attached to the shores along shallow river banks (Hicks, 2009) or adjacent to the lakeshore (Barry and Gan, 2011).

(2) Floating ice: on rivers the floating frazil ice built up to ice pans on the surface of the water by collisions; these forms are also known as pancake ice (Hicks, 2009). Floating ice can be formed on lakes as well, when the amalgamation of surface ice pans and the direct development of ice-cover are hindered by wind and waves (Barry

and Gan, 2011).



(3) Ice-cover: on rivers, when floating ice covers more than 80–90% of the surface in the profile, ice pans can be pushed and frozen together to form a solid ice-cover (Hicks, 2009). On lakes, in still conditions, ice-cover can develop directly from frazil ice (Barry and Gan, 2011).

The ice regime of rivers and lakes can be characterized by the dates of appearance and disappearance, duration and
frequency of the different ice phenomena during the winter seasons (Leppäranta, 2010; Smith, 2000). The following ice phenology records were extracted for each winter:

(1) Ice-on: when ice appears, the start date of the occurrence of floating ice

(2) Freeze-up: the start date of the occurrence of continuous, solid ice-cover

(3) Break-up: when ice-cover cracks and begins to float again, i.e. the date following the last ice-covered day

(4) Ice-off: the date of ice disappearance, the date after the last day when floating ice is present

(5) Duration of ice-cover: the number of days, when ice-cover occurred

(6) Duration of ice-affected season: the number of days when floating ice or ice-cover occurred

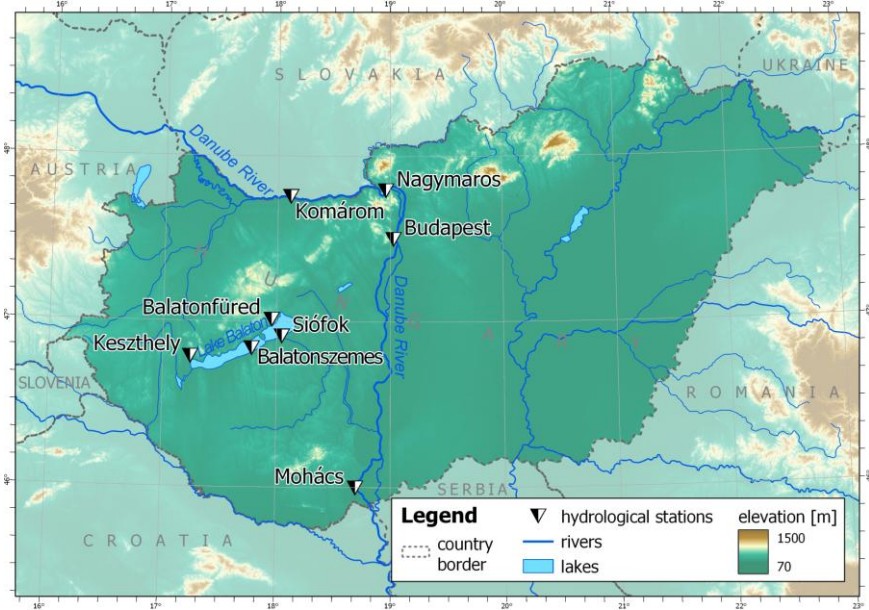

**Figure 1: Location of the hydrological stations mentioned in this study.**

## 2.2 Data sources

There are relatively a few stations around the Globe with records of freshwater ice phenology reaching back more than 100 years (Prowse et al., 2007b). However, Hungary is in a relatively good position regarding the available freshwater ice phenology data, owing to a tradition of ice observation stretching back 140 years. In East Central Europe knowledge of river



and lake ice regimes has always been important in relation to the practical requirements of fishing, transportation (navigation and crossing possibilities of rivers and lakes) or river ice jam floods (Herman, 1887; Lukács, 1934; Rácz, 2016).

There are three major sources of freshwater ice phenology:

    (1) regular monitoring

(2) historical observations: relating to shorter periods

    (3) documentary sources

(1) In Hungary regular, daily observations started in 1876, simultaneously with construction and installation of the water level monitoring network (VITUKI, 1974). In the beginning, monitoring was related to only ice-cover; no records were kept of floating ice (Lászlóffy, 1934). In this study, 4 stations were selected along the Danube which had been active right from

the beginning of regular ice observations: Komárom, Nagymaros, Budapest, and Mohács (VITUKI, 1974; Figure 1). Regular ice monitoring at Lake Balaton began only in 1925, at Siófok (Figure 1), and a network later developed with the addition of other stations (Baranyi, 1975). A few occasional observations on the ice phenology of Lake Balaton were also available from the Hydrographic Yearbooks dating back to 1885.

From 1876, a summary of freshwater ice observation data was published together with daily water level records in the

Hydrographic Yearbooks. Freshwater ice phenomena were recorded on daily basis (i.e. day-month-year) and the defining criteria of the different freshwater ice phenomena remained common and consistent through the entire monitoring period, so the dataset is homogenous from a methodological point of view (Assel and Herche, 1998), and is characterized by a high degree of precision in its temporal resolution. Recent observations (from 2000) and digitalized former records are available from the Hungarian Hydrological Database (MAHAB; Klausz and Pászthory, 2001). Data collection focusing on river ice

observations 1900–1970 have also been published in a separate book including processed data for 117 stations on several different rivers (VITUKI, 1974).

(2) Historical freshwater ice observations relating to shorter periods are available for the Danube and Lake Balaton as well. These records are suitable for extending the existing time series of observations back to a date preceding the beginning of regular river and lake ice monitoring. Periodical observations were gathered and digitalized to complete the time series of

regular freshwater ice observations on Danube River and Lake Balaton (Table 1). Although, historical observations have not been taken at the same place on the lake shore (Figure 1), it was demonstrated that when multiple observation records were compared around the lake no significant difference could be found in the ice regimes recorded at different places around Lake Balaton (Starosolszky, 1988). So, to get the most complete and longest time series of the lake ice regime for Balaton, all records were stacked. For sake of simplicity hereinafter the stacked record of Lake Balaton will be referred to as

"Balaton".

(3) From the 16 and 17$^{\text{th}}$ centuries several written data sources are to be found which relate to freshwater ice phenomena, although systematic observations from that period are not available (Vadas, 2013). Based on the most comprehensive collection of documentary evidence on natural disasters and calamities in the historical area of Hungary (Réthly, 1970; Réthly and Simon, 1998, 1999), some information about freshwater ice regimes may be gathered. The validity and accuracy





of the 19<sup>th</sup> century ice phenomena records before the beginning of the official monitoring is supported by the excellent correspondence between the seasonal removal and re-establishment of the pontoon bridge regularly used before the construction of the first stone bridge in the Hungarian capital (Rácz, 2016).

The long-term dataset of freshwater ice observations was compiled based on above mentioned data sources for the following periods:

(1) River Danube: Komárom 1876–2017, Nagymaros 1876–2017, Budapest 1774–2017, Mohács 1876–2017

(2) Lake Balaton: 1885–2017

| water body | period | reference |
|---|---|---|
| River Danube | 1847–1850 | Arenstein, 1850 |
| River Danube | 1851–1861 | Fritsch, 1864 |
| River Danube | 1860–1862 | Fritsch, 1867 |
| River Danube | 1817–1932 | Kuzmann, 1981; Lászlóffy, 1934 |
| Lake Balaton | 1892–1904 | Cholnoky, 1907; Sáringer, 1900 |
| Lake Balaton | 1920–1934 | Lukács, 1934 |

**Table 1: Additional data sources concerning the ice regime of the River Danube and Lake Balaton.**

### 2.3 Data pre-processing

For the statistical evaluation and long-term trend detection, the raw version of freshwater ice phenology records underwent some pre-processing.

(1) The calendar dates of ice phenomena appearances and disappearances were converted to numbers. November 1<sup>st</sup> was designated as 1 and the other dates were assigned numbers up to 160, that is, April 10<sup>th</sup>.

(2) Because of the climatic conditions of the investigated area, ice phenomena can appear and disappear more than one time in the course of the same winter. The first appearances (ice-on and freeze-up) and final disappearances (break-up and ice-off) were therefore selected from the station records for further analysis.

(3) The duration of the ice covered and ice-affected seasons were calculated taking into account the mid-winter break-up or ice free periods.

(4) A separate analysis was carried out of winter records in which no ice phenomena were observed.

### 2.4 Analysis of the long-term trends in the East Central European freshwater ice phenology records

A temporal analysis of the ice phenology records was carried out to quantify the long-term changes in freshwater ice phenomena on the Danube and Lake Balaton. The linear trend was evaluated by applying two methods: (1) Linear





regression, a parametric method for trend testing in which the trend magnitude can be estimated from the regression slope, and (2) A non-parametric alternative for trend testing, the Mann-Kendall test (Kendall, 1975; Mann, 1945), in which the trend magnitude may be calculated with the use of the Sen slope estimator (Sen, 1968). The advantage of the second method is that it is not sensitive to missing values. Trend estimations are related to the period 1885–2017 in Sections 4.1 and 4.2

below. In all cases, the magnitude of trend was expressed in days/100 years. A positive trend means that the date of ice phenomena appearance/disappearance has shifted to later dates, or the duration of ice phenomena has increased, while a negative trend indicates the earlier appearance/disappearance of ice phenomena, or the decreasing duration of ice phenomena.

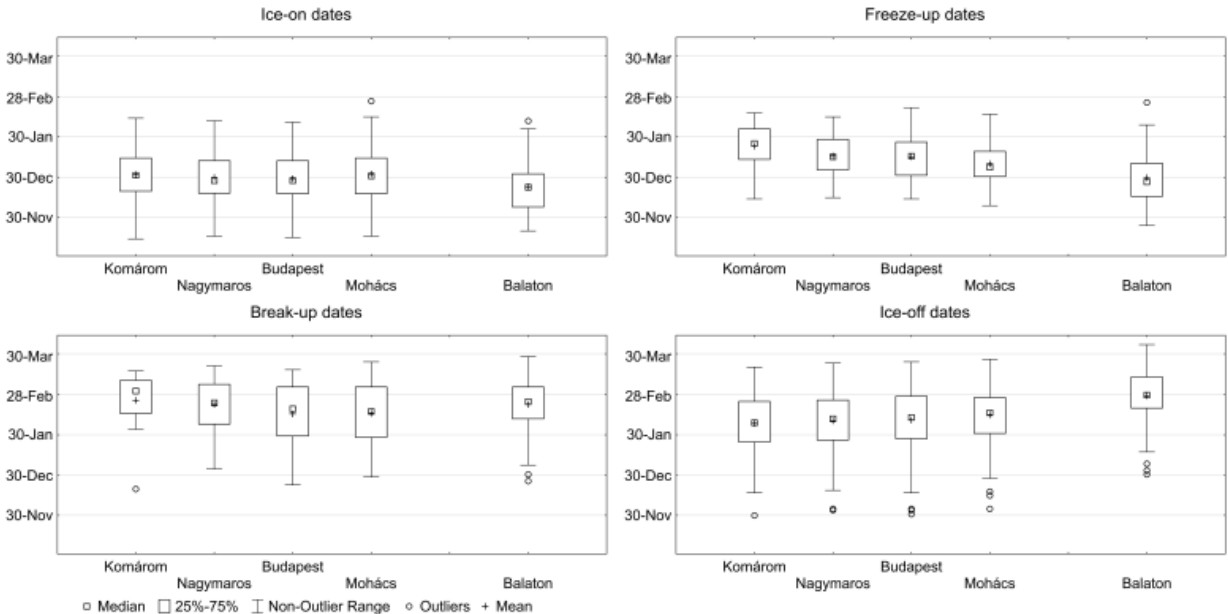

**Figure 2: Basic characteristics of the ice regime of the River Danube and Lake Balaton 1885–2017.**

**3 Characteristics of the ice phenology**

Descriptive statistics of ice phenology records compared for the longest common period (1885–2017) when freshwater ice observations for all stations were available (Figure 2). The longest observation dataset (of the River Danube at Budapest) was also tested to investigate the differences and changes in the occurrence of ice phenomena compared to the shorter, recent

period.

**3.1 The River Danube**

Ice occurrence is in general to be expected between late December and late February; ice cover usually builds up from mid-January to mid-February over the investigated section of the Danube, but in extreme cases these dates can vary as far as





covering the period from November to March (Figure 2). On the basis of the observations, the earliest date of ice-on was 14th November 1888, and the latest date of ice-on was 26th February 1986. The earliest and the latest freeze-up times were recorded on 9th December 1925 and on 21st February 1887, respectively. In the case of break-up, the earliest date was 20th December 1902, while the latest was 26th March 1895. The earliest observed ice-off date was 30th November 1915, while the

latest was 28th March 1940. In case of the extended observations from Budapest, the mean and extreme dates are similar, earlier ice-on and freeze-up were only recorded on 11th November 1876 and 7th December 1774, respectively.

Considering the entire section of the Danube, ice-on occurs earlier at the upper stations, because of the ice-drifting arriving from upstream. However, freeze-up starts at the lower sites and then the ice-cover is built up in an upstream direction. Break-up also occurs earlier in the direction of flow, but by way of contrast, ice-off takes place later on the lower sites. There

is an apparent contradiction between the mean dates of break-up and ice-off, so that the break-up occurs later than the ice-off on average. The explanation for this odd feature is the presence of winters with no ice-cover at all (only drift-ice appears). If ice-cover is formed, the date of ice-off is to be found much later compared to years without any ice-cover formation (Table 2).

The interannual variability of ice phenomena based on the standard deviation of the dates for the appearance of ice

phenomena and their disappearance are similar at all of the observation sites along the investigated section of Danube. In the case of ice-on and freeze-up, this is 15–21 days, while for break-up and ice-off it is somewhat higher, 20–25 days.

Over the investigated section of the Danube, the length of the ice-affected season – excluding the ice-free winters – is 26–35 days on average, which increases in the direction of flow (Figure 3). The longest ice-affected season was observed in 1894/1895, when 100 days of ice phenomena were recorded. The duration of ice-cover is 29–36 days on average (omitting

winters without ice-cover), which also leads to an apparent contradiction (Figure 3). Here, the explanation is similar, with the shift in mean statistics being caused by the presence of ice-cover free winters, when the duration of the ice-affected season is much shorter than when ice-cover develops as well. The longest duration of ice-cover was recorded in 1946/1947, when ice-cover endured for 83 days. The extended historical observation dataset of Budapest contains only one longer record, in 1829/1830, when the ice-cover lasted for 99 days. The interannual variability of the ice-covered and the ice-affected season

is 20–23 days and 22–28 days, respectively.



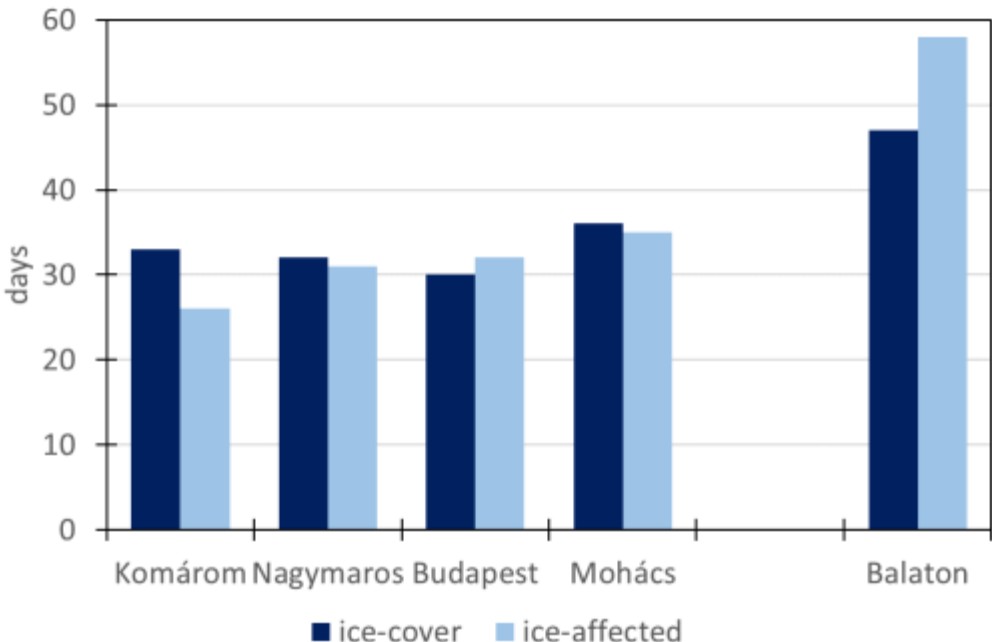

Figure 3: The average length of the ice-covered and ice-affected season of the Danube and Lake Balaton 1885–2017.

| station | break-up | ice-off (whole) | ice-off (ice-cover) |
|---|---|---|---|
| Komárom | 24-Feb | 07-Feb | 02-Mar |
| Nagymaros | 21-Feb | 09-Feb | 01-Mar |
| Budapest | 14-Feb | 09-Feb | 27-Feb |
| Mohács | 15-Feb | 13-Feb | 26-Feb |
| Balaton | 21-Feb | 28-Feb | 28-Feb |

Table 2: Comparison of break-up and ice-off dates relating to the whole dataset (whole) and winter with ice-cover (ice-cover) on the Danube and Lake Balaton 1885–2017.

### 3.2 Lake Balaton

Ice-on and freeze-up can occur as early as late December on Lake Balaton, while the usual period of break-up and ice-off usually take place just in late February and at the end of February, respectively (Figure 2). Based on long-term ice observations, the earliest ice-on was recorded on 20[th] November 1956, and the latest ice-on was observed on 11[th] February 1975. The earliest date of freeze-up was on 24[th] November 1914, while the latest was on 25[th] February 1902. In the case of break-up, the earliest and the latest records were 26[th] December 1973 and 30[th] March 1929, respectively. Finally, the earliest date of ice-off was observed on 31[st] December 1926 and the latest ice-off date was recorded on 8[th] April 1940. The



interannual variability of the dates of the appearance and disappearance of lake ice are 19–22 days in case of the Lake Balaton, which is close to that of the Danube.

The length of the ice-affected season on Lake Balaton – excluding the ice-free winters – is 58 days on average, which is 2–3 weeks longer than in the case of the Danube (Figure 3). The longest ice-affected season was observed in 1962/1963, when
118 days with ice phenomena were recorded. The duration of ice-cover is also longer, 47 days on average (discounting winters with no ice-cover; Figure 3). The longest duration of ice-cover was observed in 1969/1970, when ice-cover remained for 110 days. Comparing to the River Danube, ice-cover is a more frequent phenomenon on Lake Balaton, with only 5 winters in which no ice-cover developed after ice-on since 1885. The interannual variability of the ice-covered and the ice-affected season is 25 and 28 days, respectively, similar to that of the Danube.

**4 Long-term trends in East Central European freshwater ice phenology records**

**4.1 The River Danube**

On the basis of the observations, ice-on dates have shifted to a significantly later date over the entire section of the Danube. Similarly, freeze-up dates have also appeared later, but this change was not significant for all stations. In the case of break-up, some differences could be detected: at Komárom and Nagymaros practically no changes could be found, but at Budapest
and Mohács break-up has shifted to an earlier point in time, although the trend magnitude was not significant (p>0.05). However, in the last 50 years, ice-cover has occurred only once on this river section. Ice-off has also appeared consistently earlier, and in this case the change was significant (Figure 4, Figure 5). As a result of later freeze-up and earlier break-up, the duration of ice-cover has decreased.

An exception is Komárom, where a slight, but not significant increase could be detected. Due to later ice-on and earlier ice-
off, the ice-affected season significantly shortened at all sites on the River Danube (Table 3).

The number of ice free and ice-cover free winters has increased over the investigated period on the Danube, and especially from the 1970s. Between 1885 and 1965 between 5–9 ice-cover free winters occurred per decade at Komárom and Nagymaros, and 2–7 at Budapest and Mohács, but after 1975 the number of ice-cover free winters rises to 9–10. Ice free winters have also become more frequent, again, especially since 1975.

During the analysed period anthropogenic interventions have intensified remarkably on the Hungarian section of Danube and also upstream, and it could be this which also affects the river ice regime, besides the more general phenomena of climate change and increasing winter temperatures (Takács et al., 2013). In the case of the Danube, river regulation works and increasing water pollution have affected the river ice regime. The changes were the most conspicuous after the 1960s, when the relative frequency and the duration of ice phenomena and winter air temperatures triggering ice occurrence significantly
decreased compared to the previous periods (Takács et al., 2013). So these anthropogenic effects, together with increasing temperatures, may well be the cause of the long-term trends in the river ice regime of the Danube River.



| [days/100 years] | | ice-on | freeze-up | break-up | ice-off | ice-cover | ice-affected |
|---|---|---|---|---|---|---|---|
| | | | | Danube River | | | |
| Komárom | R | **+28.28** | +4.79 | +1.00 | **-17.91** | +1.18 | **-25.74** |
| | MK | **+30.30** | +4.35 | 0.00 | **-21.30** | +1.76 | **-22.97** |
| Nagymaros | R | **+19.96** | +7.62 | +4.06 | **-23.19** | -16.64 | **-34.06** |
| | MK | **+21.25** | +14.94 | 0.00 | **-23.87** | -15.79 | **-30.61** |
| Budapest | R | **+20.84** | +7.35 | -10.65 | **-25.68** | -10.04 | **-33.48** |
| | MK | **+21.98** | +12.50 | -7.35 | **-27.06** | -8.16 | **-29.27** |
| Mohács | R | **+23.87** | +11.73 | -8.42 | **-25.72** | -15.11 | **-42.48** |
| | MK | **+22.95** | +12.14 | -12.50 | **-26.19** | -16.99 | **-41.40** |
| | | | | Lake Balaton | | | |
| Balaton | R | **+11.56** | +6.56 | +0.67 | *-9.04* | -0.73 | -8.56 |
| | MK | **+13.33** | *+9.09* | 0.00 | **-9.18** | 0.00 | -10.53 |

**Table 3: Changes in the ice regime of the River Danube and Lake Balaton 1885–2017. (Values with p < 0.05 are shown in bold, and p < 0.10 are given in italic; R=trend calculated by linear regression, MK= trend calculated by Mann-Kendall test).**



**Figure 4: Variations in the dates of ice-on and ice-off on the River Danube 1885–2017. (Data was smoothed using 10 year moving averages.)**







**Figure 5: Variations of the date of freeze-up and break-up on the River Danube 1885–2017. (Data was smoothed using 10 year moving averages.)**



### 4.2 Lake Balaton

On the basis of the observations, ice-on (11.56 days/100yrs, $p<0.05$) and freeze-up (6.56 days/100yrs, $p>0.1$) have shifted to later dates on Lake Balaton. The magnitude of trend in ice-on was less than a half that of the Danube, while in the case of

5 freeze-up, it is about the half of that. The date of break-up remains practically unchanged, but ice-off has moved to a significantly earlier timing. The ice-off trend magnitude was also less than half that of the Danube (Figure 6). The duration of ice-cover has not changed, either, with only slight changes in freeze-up and break-up dates. The ice-affected season has shortened as a result of later ice-on and earlier ice-off dates, as on the Danube, but the magnitude of this change was much lower than on Danube (Table 3).

The frequency of ice free or ice-cover free winters has not changed over the analysed period at Lake Balaton. Compared to the Danube, ice free winters were very rare at Lake Balaton; only 0–2 winters with no ice-cover and 0–1 winters with no ice have occurred per decade since 1885.

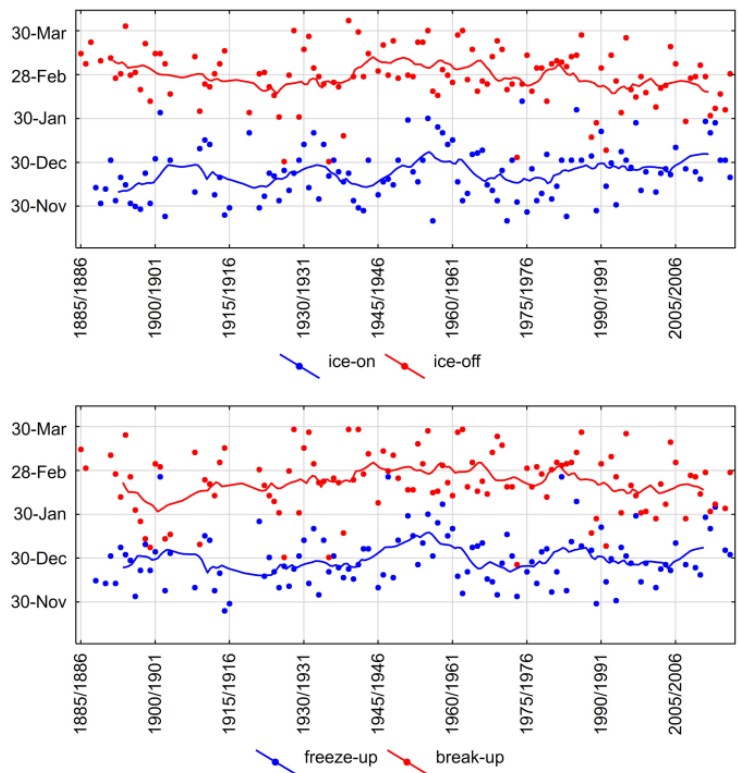

**Figure 6: Variations of the date of ice-on, freeze-up, break-up and ice-off on Lake Balaton 1885–2017. (Data was smoothed using 10 year moving averages.)**



| [days/100 years] | Danube, Budapest (1875–2017) this study | Drava, Barcs (1875–2014) Takács and Kern, 2015 | Raba, Szentgotthárd (1875–2014) Takács et al., 2013 | Vistula, Toruń (1861-2003) Pawłowski 2009* |
|---|---|---|---|---|
| ice-on | **+21.45** | **+22.22** | +6.96 | +19 |
| freeze-up | +10.20 | +4.41 | +5.75 | +13 |
| break-up | -10.24 | -11.33 | *-16.13* | -8 |
| ice-off | **-23.08** | **-12.38** | **-10.07** | -11 |
| ice-cover | -14.62 | -9.02 | *-14.93* | -39 |
| ice-affected | **-28.24** | **-21.92** | **-22.41** | -34 |

**Table 4: Changes in the ice regime of the Rivers Danube Drava and Raba. (Values with $p < 0.05$ are shown in bold, and $p < 0.10$ are given in italic.) *: statistical significance was not published in the original study**

### 4.3 Regional and global outlook

River ice regime trends relying on an observation period exceeding 100 yrs have been investigated in the case of some other water bodies in East Central Europe (Pawłowski, 2009, 2015; Takács, 2016; Bączyk and Suchożebrski, 2016; Takács and Kern, 2015), providing an opportunity for regional comparison. The investigated trends compared to the trend results of this study display a similar direction, but are of a different magnitude (Table 4). On the Danube, the trend of the ice-on date is similar to the changes detected on the Drava, but much faster than the Raba. Freeze-up dates and ice-off dates on the Danube have changed twice as fast as on the Drava and Raba. The duration of ice-cover has decreased to a similar degree on both the Danube and Drava, and somewhat faster than on the Raba. The length of the ice-affected season has also changed the most on the Danube. Only in the case of break-up could there be found slower changes on the Danube than on the Drava or Raba (Table 4).

A recent study evaluated the ice regime trends of the Lake Balaton over a shorter period (1926–2013) utilizing only the records available from the Central Transdanubian Water Authority, Székesfehérvár, Hungary. The results relating to the trends in ice phenomena were contrary to those observed in the present study: it was found that freeze-up has not changed, break-up has shifted earlier by 7 days/100 years, and the duration of ice-cover has decreased by 12 days/100 years (Soja et al., 2014). This discrepancy calls attention to the importance of a common reference period in trend analysis and climate research.

However, long-term freshwater ice phenology data are available mainly for stations outside of East Central Europe (Benson and Magnuson, 2012). A detailed evaluation of the large-scale climatic information potentially retrievable from the freshwater ice phenology records of the Danube and Balaton is beyond the scope of this study; it was, however, decided to present a graphical comparison with the longest similar records from Asia, North America and Europe to highlight the scientific value of these new cryophenological records.





**Figure 7: Variations in the dates of freeze-up and break-up on The Red, Tornionjoki, Angara Rivers and River Danube 1774–2017. (Data were smoothed using 10 year moving averages. Trend magnitudes are expressed in days/100 years and are marked with the same color.)**





**Figure 8: Variations in the dates of freeze-up and break-up on Lakes Mendota, Näsijärvi, Baikal, Suwa and Balaton 1885–2017. (Data were smoothed using 10 year moving averages. Trend magnitudes are expressed in days/100 years and are marked with the same color.)**



For the period of 1774–2017 the changes in freeze-up dates, break-up dates and duration of ice-cover of the Danube were compared to the Red River (Canada-USA), the Angara (Russia), and Tornionjoki (Finland-Sweden). To eliminate the bias of anthropogenic interventions trend magnitude was calculated for the 1774–1960 period in the case of the Danube for the sake of comparison. The dates of freeze-up have shifted to later dates in all cases, but on the Danube and Angara the changes were slower than on the Red River. The dates of break-up have shifted to earlier dates except in the case of the Angara River, where later break-up was observed. The magnitude of trend in the Danube is lower compared to the Red and Tornionjoki Rivers. The duration of ice-cover decreased on the Danube and Red Rivers, but on the Angara no significant changes could be detected (Figure 7).

Freeze-up dates, break-up dates and the duration of ice-cover at Lake Balaton were compared to the corresponding records for Lake Mendota (USA), Lake Baikal (Russia), Lake Suwa (Japan), and Nasijarvi (Finland). In the period 1885–2017 the dates of freeze-up have shifted to later date at all lakes, but the magnitude of the trend was lower at Lake Balaton than the other lakes. In the case of break-up dates, earlier shifts were observed, but no significant change was detected in the case of Lake Balaton. The duration of ice-cover has not changed either at Balaton, but at the other lakes decreasing trends have been detected in both instances (Figure 8).

## 5 Conclusions

Centennial records of the freshwater ice regime of the largest river (Danube) and the largest lake (Balaton) in East Central Europe were compiled. Based on observations covering the period 1885–2017 for the River Danube and Lake Balaton, the freshwater ice regime has significantly changed. On the Danube the dates of ice-on and freeze-up have shifted to later dates by 21–30 and 4–15 days/100 years, respectively, while on Lake Balaton these changes were only 13 and 9 days/100 years. Break-up dates and ice-off dates have shifted earlier by 7–13 and 21–27 days/100 years on the Danube, but at Balaton the date of break-up has not changed significantly, while ice-off has moved later by only 9 days/100 years. The changes in the dates of ice phenomena have resulted in a shortening of both the ice-covered and ice-affected periods. The duration of the ice-affected season has decreased on both water bodies by 23–41 days/100 years on the Danube and 11 days/100 years on Balaton. The duration of ice-cover has decreased by 8–17 days/100 years on the Danube, while on Lake Balaton no significant changes could be detected. Comparing on regional and global scales, the detected temporal trends in freshwater ice regimes are congruent with the changes in other water bodies for which long-term time series of ice phenology records are available. The changes are in the same direction, but of different magnitude.

These long-term time series of freshwater ice regimes represent a great potential resource for (paleo)climatological research due to the strong, physically deterministic link between water and air temperature and freshwater ice phenomena. The compiled cryophenological records for East Central Europe are readily available for detailed analysis of the temporal trends, large-scale spatial comparison or other climatological research purposes.



**Data availability**: The derived dataset is publicly available via PANGAEA at doi.pangaea.de/10.1594/PANGAEA.881056

**Author contribution**: KT collected the data and performed the statistical analyses. KT and ZK interpreted the results with contributions of LP. KT and ZK drafted the manuscript which further improved with contributions from LP.

**Competing interest**: The authors declare that they have no conflict of interest.

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
