# Peer review of "Long-term ice phenology records from East Central Europe"

_Earth System Science Data, 2017_

## Referee Comment (RC1) · C. Gries (Referee) · 21 Nov 2017

This is an important dataset of ice phenology data for the river Danube and the Lake Balaton. It will complement many other such datasets and allow for better analyses of the variations in ice phenology across the Northern Hemisphere. Many such data have been gathered by many authors and used extensively, however, every fresh water body behaves differently and large scale conclusions need to be based on datasets that account for this variation. Hence, this dataset will make an important contribution to this research area.

Since this paper only introduces the dataset there is not much more to discuss. Only very basic analyses have been done alluding to a different paper that goes into more

depth analyzing these data in context.

The section 3 'Characteristics of the ice phenology' is a bit tedious to read and I think a table could help make a little more sense of all those dates.

Detailed edits:

Page 3 line 17: There are relatively few stations (delete 'a')

Page 4 line 15: on 'a' daily basis

Page 4 line 24: the word is digitized not digitalized

Page 4 line 29: For 'the' sake of simplicity 'hereafter' ..

Page 9 line 12: 'On' the basis

---

## Referee Comment (RC2) · B. Pawłowski (Referee) · 13 Dec 2017

Several issues in the article require, in my opinion, minor corrections or additions before publishing. The most important are listed below:

Ice phenology data. Other causes of the formation of ice cover on rivers are also possible. For example, border ice zones can merge, and besides freezing over as a result of abundant presence of pancake ice forms, their flow can be locally restrained, forming ice bridges. Moreover, ice floes – the form of ice occurring during ice cover break-ups – are not mentioned, either. Consider replacing the terms 'ice-on' and 'ice-off' used for the beginning and the end of ice phenomena with others, more common in scientific literature. Perhaps it would be useful to mention ice jams on the river, if there

is available data. Are they less frequent nowadays, or do they occur at a later date than before? This kind of information should considerably improve the value of the paper.

Data pre-processing. Some of the statements provided in the methods section are obvious to me and need not be explained. This, for example, applies to the information on the conversion of dates into numbers to determine the mean dates, or the explanation regarding the interpretation of a positive trend.

Long-term trends in East Central European freshwater ice phenology records. In the case of the Lower Vistula River section the duration of ice phenomena was found to be correlated with the pollution of the river water, especially as regards changes in the (annual mean) concentration of chlorides, which in the years 1960-2014 increased from approx. 40 to 200 mgÅůdm–3 (Pawlowski B., 2017, Course of ice phenomena on the Lower Vistula River in 1960-2014, UMK Torun, 176 - in Polish, summary in English). Perhaps similar information for the Danube could be provided as well? As regards the Lower Vistula, in 1960-2016 the duration of ice phenomena substantially decreased: the changes ranged from 0.66 to 0.96 daysÅůyear–1 (the values were even higher and in the case of the gauging station located directly downstream of the Włocławek Dam). It was estimated that a change in the water chemistry could be accountable for up to 25-30% of the changes referred to above. Perhaps it could be useful to focus on anthropogenic factors which could have contributed to the reduction of the time the ice phenomena occur on the river. In my opinion some information on the hydro-engineering infrastructure in the area of the described stations on the Danube should be included. River control measures result in a delayed development of ice cover, its shorter holding period and a lower frequency of occurrence. In the case of the lowest section of the Vistula River the ice cover holding period is also reduced because of ice-breaking actions.

---

## Author Comment (AC1) · 24 Jan 2018

**First of all we say thanks to Dr Corinna Gries for her time spent on the review of our manuscript. We appreciate that she evaluated the presented data as interesting. Below we provide replies to her questions typed in boldface.**

This is an important dataset of ice phenology data for the river Danube and the Lake Balaton. It will complement many other such datasets and allow for better analyses of the variations in ice phenology across the Northern Hemisphere. Many such data have been gathered by many authors and used extensively, however, every fresh water body behaves differently and large scale conclusions need to be based on datasets that account for this variation. Hence, this dataset will make an important contribution to this research area.

Since this paper only introduces the dataset there is not much more to discuss. Only very basic analyses have been done alluding to a different paper that goes into more depth analyzing these data in context.

**We appreciate that our Reviewer evaluated the presented data as interesting.**

The section 3 'Characteristics of the ice phenology' is a bit tedious to read and I think a table could help make a little more sense of all those dates.

**The suggested table has been added to Section 3 summarizing the earliest, mean, and latest records of the studied ice phenomena on the Danube and Lake Balaton**

|         |           | earliest date | mean date | latest date |
|---------|-----------|---------------|-----------|-------------|
| Danube  | ice-on    | 14-Nov        | 30-Dec    | 26-Feb      |
|         | freeze-up | 09-Dec        | 14-Jan    | 21-Feb      |
|         | break-up  | 20-Dec        | 19-Feb    | 26-Mar      |
|         | ice-off   | 30-Nov        | 10-Feb    | 28-Mar      |
| Balaton | ice-on    | 20-Nov        | 23-Dec    | 11-Feb      |
|         | freeze-up | 24-Nov        | 30-Dec    | 25-Feb      |
|         | break-up  | 26-Dec        | 21-Feb    | 30-Mar      |
|         | ice-off   | 31-Dec        | 28-Feb    | 08-Apr      |

Detailed edits:
Page 3 line 17: There are relatively few stations (delete 'a')
**corrected**
Page 4 line 15: on 'a' daily basis
**corrected**
Page 4 line 24: the word is digitized not digitalized
 **corrected**
Page 4 line 29: For 'the' sake of simplicity 'hereafter' ..
**corrected**
Page 9 line 12: 'On' the basis
**Thanks for the corrections of these mistakes.**

---

## Author Comment (AC2) · 24 Jan 2018

**First of all we say thanks to Dr Bogusław Pawłowski for his time spent on the review of our manuscript. Below we provide replies to her questions typed in boldface.**

Several issues in the article require, in my opinion, minor corrections or additions before publishing.

The most important are listed below:
Ice phenology data. Other causes of the formation of ice cover on rivers are also possible. For example, border ice zones can merge, and besides freezing over as a result of abundant presence of pancake ice forms, their flow can be locally restrained, forming ice bridges. Moreover, ice floes – the form of ice occurring during ice cover break-ups – are not mentioned, either.
**Ice bridges formed by the merging of border ice zones and ice floes have been mentioned in the revised 2.1 Ice phenology data section.**

Consider replacing the terms 'ice-on' and 'ice-off' used for the beginning and the end of ice phenomena with others, more common in scientific literature.
**We think the terms 'ice-on' and 'ice-off' is quite common in scientific literature. As an argument we can say that 5 out of the 21 cited English-language references which discuss freshwater ice phenomena use these terms; including the seminal paper of Magnuson et al., 1999 and one of the latest review articles on lake ice trends by Weyhenmeyer et al., 2011.**
**By a way of contrast only a single study uses the terminology of 'first ice/ice disappearance'.**
**(The other 15 deals only with freeze-up and/or break-up, or with a different aspect of freshwater ice phenology.)**
**An advantage of the used terminology 'ice-on/ice-off' is its brevity compared to other more verbose expressions. In addition, we define the meaning of these compact expressions in the 2.1 Ice phenology data section so it should be clear for the Readers. If the Editor does not say otherwise we are to keep the original terminology.**
**Magnuson et al. 1999 Historical Trends in Lake and River Ice Cover in the Northern Hemisphere, Science, 289(5485), 1743–1746, doi:10.1126/science.289.5485.1743**
**Weyhenmeyer et al., 2011 Large geographical differences in the sensitivity of ice-covered lakes and rivers in the Northern Hemisphere to temperature changes, Glob. Chang. Biol., 17(1), 268–275, doi:10.1111/j.1365-2486.2010.02249.x**

Perhaps it would be useful to mention ice jams on the river, if there is available data. Are they less frequent nowadays, or do they occur at a later date than before? This kind of information should considerably improve the value of the paper.
**There were certain well documented ice jams especially on the Danube. They have caused devastating ice floods. However, the ice jams are not recorded systematically in the hydrological year books, or their main occurrence did not happened at stations with the presented long ice phenology records.**
**Their compilation would require a separate project collecting historical hydrological evidence from additional documentary sources, or select other station(s), maybe having shorter record on ice phenology but more complete record on ice jams. We feel it is out of the scope of the current study.**

Data pre-processing. Some of the statements provided in the methods section are obvious to me and need not be explained. This, for example, applies to the information on the conversion of dates into numbers to determine the mean dates, or the explanation regarding the interpretation of a positive trend.

**We accept that some of the methodological details seem to be trivial to an expert, however we reckon that they might be helpful for interested Readers&Users from other disciplines. Therefore, we think the detailed documentation is useful (and essential for a data paper) and wish to keep this part of the Methods section.**

Long-term trends in East Central European freshwater ice phenology records. In the case of the Lower Vistula River section the duration of ice phenomena was found to be correlated with the pollution of the river water, especially as regards changes in the (annual mean) concentration of chlorides, which in the years 1960-2014 increased from approx. 40 to 200 mg dm–3 (Pawlowski B., 2017, Course of ice phenomena on the Lower Vistula River in 1960-2014, UMK Torun, 176 - in Polish, summary in English). Perhaps similar information for the Danube could be provided as well?

As regards the Lower Vistula, in 1960-2016 the duration of ice phenomena substantially decreased: the changes ranged from 0.66 to 0.96 days year–1 (the values were even higher and in the case of the gauging station located directly downstream of the Włocławek Dam). It was estimated that a change in the water chemistry could be accountable for up to 25-30% of the changes referred to above. Perhaps it could be useful to focus on anthropogenic factors which could have contributed to the reduction of the time the ice phenomena occur on the river. In my opinion some information on the hydroengineering infrastructure in the area of the described stations on the Danube should be included.

River control measures result in a delayed development of ice cover, its shorter holding period and a lower frequency of occurrence. In the case of the lowest section of the Vistula

**Thanks for the detailed information on the anthropogenic effects on the Lower Vistula ice phenomena. We have incorporated this information into the revised version and included another study focussed on lakes and reservoirs (Solarski et al. 2011). We also note that the last paragraph of Section 4.1 the original manuscript already briefly discussed the anthropogenic effects, mentioning both river regulation works and increasing water pollution as potential non-natural effects on the river ice regime. To stress the importance of these circumstances it has been mentioned repetitively after the newly added paragraph about Lower Vistula and Silesian Upland water bodies.**

**We definitely share the Reviewer's opinion that these are important questions. Thus we would like to draw the attention of the Reviewer to the paper of Takács et al., 2013 which was dedicated to the anthropogenic effects on river ice regime using the records of 1875-2008 in Eastern Central Europe. This data paper submitted to ESSD, extends the timespan of Takács et al. (2013) and included additional stations (e.g. Mohács) forming a basis for further research on the topic. However this should be detailed in a separate classical research paper and not in this data paper.**

**Pawlowski B., 2017 Przebieg zjawisk lodowych dolnej Wisły w latach 1960–2014. Wydawnictwo Naukowe Uniwersytetu Mikołaja Kopernika, ISBN 978-83-231-3885-3**

**Solarski, M., Pradela, A., and Rzetala, M.: Natural and anthropogenic influences on ice formation on various water bodies of the Silesian Upland (southern Poland). Limnological Review, 11(1), 33-44, DOI 10.2478/v10194-011-0025-1 2011.**

**Takács, K., Kern, Z. and Nagy, B.: Impacts of anthropogenic effects on river ice regime: Examples from Eastern Central Europe, Quat. Int., 293, 275–282, doi:10.1016/j.quaint.2012.12.010, 2013.**